# Health Benefits of Whey or Colostrum Supplementation in Adults ≥35 Years; a Systematic Review

**DOI:** 10.3390/nu12020299

**Published:** 2020-01-22

**Authors:** Merran Blair, Nicole J. Kellow, Aimee L. Dordevic, Stephanie Evans, Julia Caissutti, Tracy A. McCaffrey

**Affiliations:** Department of Nutrition, Dietetics and Food, Monash University, Level 1 264 Ferntree Gully Road, Melbourne, VIC 3168, Australia; merran.blair@monash.edu (M.B.); nicole.kellow@monash.edu (N.J.K.); Aimee.Dordevic@monash.edu (A.L.D.); steph.evans@sky.com (S.E.); jlcaissutti@gmail.com (J.C.)

**Keywords:** health claims, whey, bovine colostrum, adults, food industry

## Abstract

Food-health claims are an important method of translating nutrition research to consumers. Whey and colostrum are thought to exert health benefits to adults, but it is unclear what measurable, objective health benefits they impart. This review aimed to identify the objective health benefits of bovine whey or colostrum-based beverages to healthy adults aged ≥35 years to substantiate a food-health claim. Seven databases were systematically searched. Eligible articles were RCTs that involved healthy adults aged ≥35 years, consuming whey or colostrum in beverage form and measuring objective health markers. Quality assessment and data extraction was conducted in duplicate. The searches identified 9943 papers and 16 were included in this review; 13 studies, reported across 15 papers, related to whey, one study to colostrum. The outcomes identified were body composition, bone mineral density, biochemical markers, such as blood glucose and lipids, and muscle strength and synthesis. Heterogeneous outcomes, high risk of bias and inconsistent findings resulted in inconclusive evidence to substantiate a food-health claim. Clearer reporting and consensus on a minimum set of objective measures would allow for more robust recommendations regarding food-health claims. Protecting consumers from misleading health claims will require collaboration between regulators, researchers, and the food industry.

## 1. Introduction

Food-health claims are an important method of translating and communicating nutrition research to consumers [1]. As health claims are a strong driver of consumer product selection [1,2], it can be financially beneficial for food manufacturers to display them on products. This comes with a level of responsibility on the part of both researchers and the food industry to ensure that scientific rigor and transparency regarding the synthesis and interpretation of the evidence do not come second to financial gain [3]. Complex nutrition messages are often over-simplified, and, when refuted by other scientists, leave consumers confused over who to believe, resulting in the erosion of trust in nutrition science [4]. Additionally, health claims can be confusing, are often misinterpreted [1], and they require a moderate level of health literacy that not all consumers possess [5]. 

In Europe, food-health claims are based on a submission of a dossier by a petitioner that is assessed by suitably qualified experts at the European Food Safety Authority (EFSA) [6]. This application is assessed based on whether a cause and effect relationship can be substantiated and the outcomes are published in the EFSA journal [7,8]. In Australia, claims are based on a system of self-substantiation, in which food manufacturers must notify Food Standards Australia and New Zealand (FSANZ) of the intended claim and have adequate evidence (systematic literature review) to support the claim [9]. However, the strength of this evidence is not independently assessed, nor does it need to be made publicly available [9].

The current review was commissioned by a dairy food manufacturing company in Australia, with the aim of marketing a powdered supplemental milk drink, fortified with whey and colostrum to adults ≥35 years. The purpose of this review was to identify objectively measured health benefits related to this product as demonstrated via randomised controlled trials, to substantiate a food-health claim, according to the self-substantiation requirements of FSANZ [9]. The use of the ≥35 years cut off was specified by the food manufacturer as a target market for their product and any food-health claims would be required to specify this age group.

Identifying a specific source of protein that provides increased health benefits has been of interest in the scientific community over recent years [10,11]. The components of whey, such as basic milk protein, appear to be biologically active beyond protein content alone [12,13], which suggests that whey consumption might have additional health benefits when compared to other protein sources. In adults aged ≥18 years, there is evidence for very modest improvements in body composition in females [14], mixed gender, overweight or obese cohorts [15], and in older people when whey protein was consumed in addition to resistance exercise [16]. Meta analyses have shown no effect of whey supplementation on circulating levels of the inflammatory marker C-reactive protein (CRP) [17], and insulin sensitivity outcomes were contradictory [18]. Small, but clinically insignificant, reductions in fasting and circulating triacylglycerol levels were reported following whey supplementation in adults ≥18 years, but no changes on other blood lipids were evident [19]. 

Bovine colostrum has a higher protein content than standard milk (13.0% vs. 3.3%) and it is designed to impart growth and immunological benefits to the calf in its first days of life [20]. Immunological benefits have also been demonstrated in adults with consumption leading to a reduction of damage to the gastrointestinal tract that is induced by high doses of non-steroidal anti-inflammatory drugs [21,22]. Colostrum supplementation might also reduce subjective upper respiratory symptoms in athletes according to a systematic review [23]. However, the strength of this evidence is hindered by a limited number of high quality studies [23]. 

Preliminary evidence suggests that consumption of either whey or colostrum might impart health benefits to adults beyond protein content alone. The specific population for this review, adults aged ≥35 years, was selected by the dairy food manufacturing company as a key target for prevention, or slowing the progression, of age-related diseases. However, outcomes were not specified prior to commencement, and the review was designed to determine what objectively measurable health benefits might exist. 

This review is the first, to the authors’ knowledge, to specifically address objective health benefits of whey or colostrum consumption in apparently healthy adults aged ≥35 years. It was conducted alongside industry, with the purpose of identifying a food-health relationship to substantiate a product health claim, in Australia, for a supplemental milk drink fortified with whey and colostrum. The funding providers, both food industry and a government grant, had no input into the research methodology or interpretation of results, and the publication of results did not require funder pre-approval. In this way, scientific integrity was maintained [3,24], while still achieving a collaborative outcome that might benefit science, the food industry, and consumers.

## 2. Materials and Methods 

For the self- self-substantiation requirements of FSANZ [9], this review was conducted in accordance with PRISMA reporting standards [25]. On 5th September 2018 Ovid Medline, Ovid Medline In-Process, PsycINFO, EMBASE, Scopus, CINAHL Plus, and Cochrane library databases were searched. The Population Intervention Comparator Outcome Study design (PICOS) format was defined as; apparently healthy adults aged ≥35 years, consuming whey or colostrum-based beverage, when compared to a control beverage, relating to undefined outcomes that were reported in randomised controlled trials. Search terms used were; (adult OR adults) AND (Colostrum OR Protein OR Proteins OR Whey OR Casein) AND (dairy OR milk), and no search limits were imposed. Appendix A lists an example search strategy. Reference lists of included studies were hand-searched in addition to other review papers relating to whey or colostrum.

All of the resultant references were imported into a systematic review screening and data extraction software program (Covidence Systematic Review Software, Veritas Health Innovation, Melbourne, Australia), which was used to screen studies and identify those meeting the pre-specified inclusion criteria. The Covidence program automatically identified and eliminated the duplicate articles. Three researchers (MB, SE, JC) conducted screening, with two researchers independently screening each abstract. Screening was conducted by title and abstract only for first pass and full text at second pass. The study authors were contacted via email if eligibility was unclear (*n =* 3) and papers were excluded if no response was received (*n =* 2). Any conflicts were resolved and consensus reached by discussion with a third researcher (TM). Table 1 details inclusion and exclusion criteria.

A data extraction table was piloted by two researchers and amendments were made as required. Data extraction was completed by one researcher (MB) and verified by a second (either JC or SE). Data tabulated included participant characteristics, funding sources, intervention details, compliance, background diet and physical activity, outcomes measured, and results.

Two researchers (MB and either SE or JC) independently conducted the risk of bias assessment of included papers using the Cochrane Risk of Bias Tool 2.0 [26]. A third researcher (SE or JC) resolved the conflicts. 

## 3. Results

After the removal of duplicates, 9943 papers were screened, with fourteen studies (reported across sixteen papers) being included in the final review (Figure 1). In two instances, two papers reported the same study; Pal et al. [27,28] reported different outcomes from the same study, as did papers by Eliot et al. [29] and Bemben et al. [30]. The majority of studies (*n =* 9, 56%) were determined to be at high risk of bias [28,30,31,32,33,34,35,36,37], five (31%) had some concerns [27,29,38,39,40] and two (13%) were designated as low risk [41,42] (Appendix A). Across all of the papers, 43% of the risk of bias assessment domains were unclear. Study heterogeneity, with regards to variability in the mean age of participants (42 years [35] to 78 years [41]), interventions provided (dose of whey protein ranged from 40 mg [38] to 60 g [31]), length of intervention (six days [39] to 18 months [42]), diversity of study outcomes measured, and disparate risk of bias assessments precluded meta-analyses. 

### 3.1. Whey Supplementation 

Thirteen studies (reported across 15 papers) investigated the impact of whey supplementation in apparently healthy adults (total *n =* 704; range *n =* 18–121 participants) aged ≥35 years, Table 2, Table 3, Table 4 and Table 5 show the details of the included studies (please note, same papers appear across multiple tables due to the different outcomes reported). The mean age of participants ranged from 42 to 78 years. Three studies only recruited male participants [29,30,33], four recruited only females [34,35,38,39], and the remaining eight included participants of both sexes [27,28,31,32,36,37,41,42], with the percentage of females ranging from 53–86%. The provided interventions were primarily whey protein isolate (WPI) (*n =* 6) [27,28,33,36,39,42], with other studies reporting the provision of non-specific whey (*n =* 4) [29,30,31,35] (one described as whey and peptides [35]), whey protein concentrate (WPC) (*n =* 2) [32,41], and milk basic protein (MBP) (the biologically active fraction of whey protein) (*n =* 1) [38]. Two studies referred to a “milk based protein matrix”, of which whey was a primary constituent [34,37].

Dosages from the protein interventions generally ranged from 20–60 g per day, with the exception of Aoe et al., who provided 40 mg of MBP per day [38]. Most of the studies (*n =* 8) reported that the overall dietary protein intake of participants, both at baseline and during interventions, was above the Recommended Dietary Allowance (RDA (All RDAs and AMDRs are based on United States recommendations [43,44])) (0.8 g (kg·day) for adults aged ≥18 years [43]) [29,30,32,33,34,37,39,42]. Additionally, three studies reported the protein intakes that were within the Acceptable Macronutrient Distribution Ranges (AMDR) [44] of 10–35% [27,28,31]. Two studies reported protein intakes in the control groups during intervention phases that were below the RDA; 0.61 g (kg·day) (United States study) [35] and 0.7 g (kg·day) [36] (Canadian study; RDA is also 0.8 g (kg·day), according to Canadian guidelines [45]). 

The majority of studies (*n =* 12) were conducted over 12–24 weeks [27,28,29,30,31,32,33,34,35,37,38,41], with the notable exceptions of one study of six days [39], one of 14 days [36], and one of 18 months [42]. The most commonly used comparator was an isocaloric carbohydrate beverage of either maltodextrin (*n =* 6) [32,35,36,37,41,42] or glucose (*n =* 2) [27,28]. Two studies also included a third arm of a soy protein comparator [32,36] and two included a third arm of casein [27,28]. Specific details of the “placebo beverage” were not provided for two studies [33,38], and two used a commercial sports drink [29,30]. Two studies did not use a comparator beverage [31,34], however, they involved multiple intervention arms (“whey” versus “whey plus resistance training”), and results from the whey only groups were treated as the intervention for the purposes of this review. 

#### 3.1.1. Bone Mineral Density after Whey Supplementation

As can be seen in Table 2, no changes to bone mineral density (BMD) that were measured by dual X-ray absorptiometry (DXA) were reported after six months [38] and 18 months [42] consumption of MBP and WPI, respectively (participants *n =* 148). Secondary analysis of BMD by Aoe et al. reported a marginal increase in the percentage of BMD (MBP 1.21% vs. placebo −0.66%, *p* = 0.046) [36]. However, this was smaller than the coefficient of variation (CV) for the DXA scanner (2.0%). 

Two papers reported biochemical and urinary markers that were related to bone resorption and bone formation, including osteocalcin and parathyroid hormone, which were unchanged over six and 18 months [38,42] (see Table 2). However, there was a lack of consistency of markers tested, with some only being tested in a single paper [42]. 

#### 3.1.2. Body Composition after Whey supplementation 

Ten studies assessed body composition (see Table 3) (*n =* 638) with DXA used by nine studies [28,29,31,34,35,36,37,41,42] and one study that used a BodPod^®^ [32]. Four studies intervened while using whey supplementation alone in the form of a whey blend [37], WPI [28,42], and WPC [32]. Four studies included a component of resistance training in addition to a whey blend [34], non-specific whey [29,31], and WPC [41]. Two included an energy deficit diet in addition to “whey and peptides” [35] and WPI [36].

##### Body Composition after Whey Supplementation Alone 

Without the addition of exercise or an energy deficit diet, whey supplementation did not influence body composition (*n =* 324). There was an increase in body mass (0.70 kg (95% CI 0.01, 0.35), *p* = 0.021) and lean tissue mass (0.45 kg (95% CI 0.06, 0.83), *p* = 0.006) with the consumption of a whey blend [37], but this was not significant when being expressed as a percentage of change in mass. The reported increase in body mass and lean tissue mass coincided with an increase in energy (kcal) intake [37] and this result was not replicated in other studies with WPI [28,42] or WPC [32].

The results suggested differences in the action of protein versus carbohydrate (maltodextrin) with respect to preservation of lean mass and loss of fat mass [32,42]. However, in Kerstetter et al. [42], this was confounded by a reduction in energy (kcal) intake in the group consuming WPI, which might have resulted in the reduction in fat mass. There was no difference in the change in lean mass after 23 weeks of WPC when compared to soy protein supplementation [32].

##### Body Composition after Whey + Energy Deficit Diets 

Energy deficit diets in addition to whey supplementation resulted in a reduction in total body mass in both control and whey intervention groups (*n =* 99) [35,36] (see Table 3). Frestedt et al. reported a greater reduction in fat mass following the intervention (“whey and peptides”) (−2.81 ± 0.38 kg) as compared with carbohydrate supplementation (−1.62 ± 0.33 kg) [35]. However, the margin of error (CV) was not reported for the DXA nor was the percentage of fat mass loss.

##### Body Composition after Whey + Resistance Training

In conjunction with resistance training, there was no difference in body composition measures between the groups who consumed either non-specific whey [29,31] or WPC [41] when compared to a carbohydrate control (or no control) (*n =* 215). The body composition outcomes (body mass (kg), BMI (kg/m^2^) and body fat percentage) were unreported in one study of a whey blend [34]. The results were not significantly different for male [29] and mixed-sex [31,41] groups, or in shorter (14 weeks) [29] and longer (six months) [41] interventions. 

#### 3.1.3. Muscle Strength after Whey Supplementation with or without Resistance Training

An increase in muscle strength was reported in studies that included resistance training in all groups (*n =* 176) whether they consumed non-specific whey [30], WPC [41], WPI [33], or soy, casein or carbohydrate control (see Table 4). Muscle strength was reduced in the absence of resistance training, even with whey blend supplementation [34]. 

#### 3.1.4. Muscle Synthesis after Whey Supplementation with or without Resistance Training

Conflicting results from a limited number of studies make it difficult to determine the effect of WPI consumption on the rates of muscle synthesis (*n =* 79) [33,36,39]. As can be seen in Table 4, measurements were highly variable across studies, with markers of muscle protein synthesis reduced in two studies [33,36], but less so when compared to the carbohydrate or soy protein groups in one study [36]. Fractional synthetic rate (the rate at which amino acids are incorporated into muscle fibres) was reduced in one study of WPI at a dose of 54 g protein over 14 days [36] and increased in another of WPI of 25 g protein over six days [39].

#### 3.1.5. Biochemical Markers of Disease after Whey Supplementation

##### Glucose, Insulin and Lipids after Whey Supplementation

No changes were reported (ranging from two weeks to 23 weeks) in fasting glucose levels after WPC [32], WPI [28,36] or non-specific whey supplementation [31], or in postprandial glucose levels after WPI [36] (*n =* 240) (see Table 5). There was no change in postprandial insulin after WPI [36], or fasting insulin after non-specific whey [31]. Fasting insulin was reduced when compared to either a maltodextrin or glucose control [28,32], but not significantly different when compared to soy protein [32] (*n =* 240). Fasting homeostatic model assessment of insulin resistance (HOMA-IR) was reduced after WPI consumption when compared to a glucose control [28], but unchanged when compared to non-specific whey as addition to resistance training [31]. Fasting TAG levels were reduced after WPI [28] and non-specific whey [31], whereas the levels remained unchanged in one study that did not report if the results were postprandial or fasting after whey and peptides consumption [35]. No consistent outcomes were observed for total cholesterol, LDL-cholesterol, or HDL-cholesterol (*n =* 186) [31,35].

##### Other Biochemical Markers after Whey Supplementation

Various other biochemical markers (ghrelin, insulin-like growth factor-1, leptin, adiponectin, c reactive protein, insulin-like growth factor binding protein-1, insulin-like growth factor binding protein-3, triiodothyronine, thyroxine, Glycerol RA, interleukin-6, and tumor necrosis factor alpha) were reported across different studies (see Table 5), however, there was a paucity of data and no consistency in the findings.

##### Urinary and Vascular Markers after Whey Supplementation

There was no negative impact on kidney function (estimated glomerular filtration rate (eGFR)) after 18 months of WPI (*n =* 121) (see Table 2) [42]. Urinary urea (a marker of protein metabolism) increased from baseline in the WPI group [42], but they did not change in the carbohydrate control group. The results were inconsistent across vascular measures (see Table 5) (*n =* 127), with one study showing no change in blood pressure when consuming non-specific whey [31] and another showing a reduction after WPI as compared to a glucose control, but no difference when compared to casein [27]. 

### 3.2. Colostrum Supplementation

Only one study was identified that assessed colostrum intake, the control was a whey protein complex and both groups undertook a resistance training protocol (*n =* 39) [40]. There were no differences between groups in the majority of outcomes, except leg press strength (see Table 4), which increased significantly more in the colostrum group [40]; and cross-linked n-telopeptides of type I collagen, which was reduced in the colostrum group as compared with the control (see Table 2). Significant increases were reported across both groups in bone mineral content, lean tissue mass, muscle thickness, and cognition [40] (see Table 2, Table 3 and Table 4). 

## 4. Discussion

The aim of this review was to identify the objective health benefits of whey or colostrum supplementation in apparently healthy adults aged ≥35 years, with the goal of identifying a food-health relationship and substantiating a food-health claim in Australia. The target population was pre-defined by the food manufacturer who originally commissioned this review, as this demographic was identified as the most likely consumers of their product. Outcomes relating to bone mineral density, body composition, metabolic biomarkers, muscle synthesis, and muscle strength were identified. While the total number of participants in this review was *n =* 704, specific outcomes were only tested on a limited number of these participants. A paucity of data, particularly in relation to colostrum, and lack of consistency regarding the type of supplement used, outcomes measured, and co-interventions, such as energy deficit diets and resistance training, have resulted in inadequate evidence to substantiate a food-health claim.

Protein is an essential nutrient for cell growth and maintenance and supplementation has been shown to be of benefit if protein intake is inadequate [46,47]. However, much like other essential nutrients (such as vitamins or minerals), intakes over and above physiological requirements do not necessarily confer greater health benefits. Data regarding vitamin supplementation in the absence of deficiency is limited [48], however, high doses of some (vitamin E, vitamin C, and beta-carotene) have been shown to increase risk of disease and mortality [49,50]. In the current review, the majority of participants had protein intakes that were already meeting the RDA and this could explain why improvements in health outcomes were not significant. Similar to other nutrients, individuals may need to start from a position of deficiency to see a positive effect of supplementation. 

This review did not find adequate evidence to support any specific benefits of whey protein above and beyond other protein. Three studies compared whey to other whole forms of protein, two with soy [32,36] and one with casein [27]. The body composition outcomes demonstrated no benefit of whey as compared to soy [32,36] or casein proteins [28]. Some individual results indicated enhanced benefit from whey, for example, Hector et al. found that MPS was reduced in all groups, in the absence of resistance training and after an energy deficit diet, but less so in the WPI group than the SPI group [36]. Reduced ghrelin, IGF-1, and IGFBP-3 were demonstrated, and increased thyroid hormones when compared to SPI [32]. In addition, improved insulin sensitivity was indicated when compared to casein [28]. Unfortunately, these results were only demonstrated in single studies making it difficult to determine the significance of these outcomes and further research is required.

The use of fortified milk as a carrier for functional food components has been investigated for the management of cardiovascular risk factors [51]. The addition of phytosterols and omega-3 fatty acids has demonstrated a reduction in plasma LDL cholesterol, in addition to reduced plasma triacylglycerol levels that are associated with omega-3 fatty acids [51]. However, these benefits are not due to the naturally occurring constituents of the milk (such as whey), but they are related to the functional components added to the milk, which have been shown to have the same affect in the absence of dairy milk [52,53]. 

The regulation of food and nutritional supplements and the management of food-health claims is complex and varies by region. In the USA, a predominantly whey and colostrum-based powdered formula (such as the product this review was based on) would be classified as a dietary supplement rather than a food, or a vitamin supplement [54]. In the European Union, dietary supplements are regulated in the same way as food and are “not intended to treat or prevent diseases in humans” [55]. In China, dietary supplements must be registered and have evidence of efficacy for food-health claims, either in animals or humans [56]. In Australia, proof of efficacy is required in humans, however, the regulatory agency does not always verify this evidence [9]. In the USA, the FDA website specifically states that, “For most claims made in the labelling of dietary supplements, the law does not require the manufacturer or seller to prove to FDA’s satisfaction that the claim is accurate or truthful before it appears on the product” [54]. It is somewhat perplexing that the standards for labelling of dietary supplements are not as strict as they are for food or for vitamin supplements, despite the fact that they are consumed in the same manner, for the same purpose of providing nutrition to the body. According to US regulation, the onus is placed on the consumer to be “safe and informed” [54] and to “learn to spot false claims” [54]. This leaves consumers in a vulnerable position as understanding the nuances of food-health claims takes a moderate level of health literacy [5] and the interpretation of claims can be challenging [2]. Increased transparency of substantiation, regulation, and monitoring in both the food and dietary supplement industry would provide greater consumer protection and prevent businesses profiteering from false food-health claims. 

Food-health claims that are related to whey, in both general and active populations, have not been substantiated by EFSA [57,58]. Insufficient evidence was found for a cause and effect relationship between whey protein and the following beneficial outcomes; increased satiety, maintenance of a healthy body weight, increased muscle mass, reduced body fat, increased muscle strength, increased endurance during strenuous exercise, increased muscle repair, or improved recovery from muscle fatigue after exercise [57]. Similar to the current review, the EFSA reported conflicting results from a limited number of studies, and variability in outcomes measured, in each domain [57]. The EFSA review was completed in 2010 and the current review was conducted to assess the evidence up to September 2018.

Australian regulation places the responsibility for food-health claims substantiation with the food manufacturer [9]. A list of pre-approved, high level (“related to a serious disease or biomarker of a serious disease”), and general level claims (“not a high level health claim”) that can be advertised on products are listed (Schedule 4) [59]. However, a submission must be made for additional health claims [9]. If the claim is considered to be high level, then it will be assessed by FSANZ, however, if a submission is for a general level health claim it is not routinely assessed, but is listed on the FSANZ website [9]. The systematic review supporting the claim does not have to be provided to FSANZ [9], and “while FSANZ administers the notification process, publication of a notification by FSANZ does not indicate acceptance, approval or validation of the relationship” [60]. If a company or individual disagrees with the food-health claim submitted and listed on the FSANZ website, they can complain to the relevant food safety authority or local government body [61]. However, the primary focus of these agencies is food safety, not deceptive food-health claims. Currently, a notification is listed on the FSANZ website for whey that promotes benefits with regard to weight loss [62]. This is in contradiction to the conclusions of both the EFSA and the current review. The evidence held by the manufacturer who has registered this food-health claim is not readily accessible to the general public. It would be unwise for other manufacturers to make such claims without being able to assess the evidence. However, they are at a disadvantage, as a competitor could be currently claiming weight loss benefits that are related to whey protein [62]. 

A limitation of the current review, and a major limitation in nutrition research, is a lack of clear reporting [63,64,65]. This results in wasted resources due to exclusion of results from SLRs, incorrect conclusions and inappropriate implementation of nutritional strategies [66]. Similar to other SLRs [14,67], many of the papers that were included in the current review were considered to have a moderate-high risk of bias due to a lack of clear methodological reporting. Across all of the papers, 43% of the risk of bias assessment domains was unclear. In addition, two papers were excluded as details such as health status of participants [68] and specifics of the intervention used [69] were not reported. The consistent use of checklists for reporting (CONSORT for randomised controlled trials [70]) might assist authors to achieve high standards of reporting and subsequently assist SLR authors to fully assess the potential study bias [66]. 

A further limitation of the current review was the wide variability in the biomarkers reported and the heterogeneous nature of the study designs. We would further suggest that, when examining a food-health relationship, there is consensus on a minimum set of objective measures reported across studies to permit more robust meta-analyses [66], as many are limited by the heterogeneity of included studies [15,16,71]. 

Substantiating food-health relationships is an increasing area of research and an important method for dissemination of nutrition science findings into real world applications. It falls upon the food industry to fund these projects due to diminishing funding for medical research, as they bear the legal responsibility to prove food-health claims and have the most to gain financially. Research professionals must partner with industry in order to ensure research is conducted by suitably qualified personal with adequate scientific rigor and correct interpretation of results. Maintaining public trust in nutrition research is essential and this requires that research be conducted transparently, in the spirit of openness and reproducibility. Regulators, researchers, and industry must work together to ensure the dissemination of nutritional research is benefiting consumers.

## 5. Conclusions

The evidence from this review is inconclusive as to whether there are objective health benefits of whey or colostrum supplementation in apparently healthy adults aged ≥35 years. There exists a lack of relevant studies and inconsistency in the reported outcomes. Increased reporting standards and a consensus on a minimum set of objective measures to substantiate food-health relationships would improve the applicability of nutrition research in the area of food-health claims. Protecting consumers from misinformation and false claims should be a priority for regulators, researchers, and the food industry.

## Figures and Tables

**Figure 1 nutrients-12-00299-f001:**
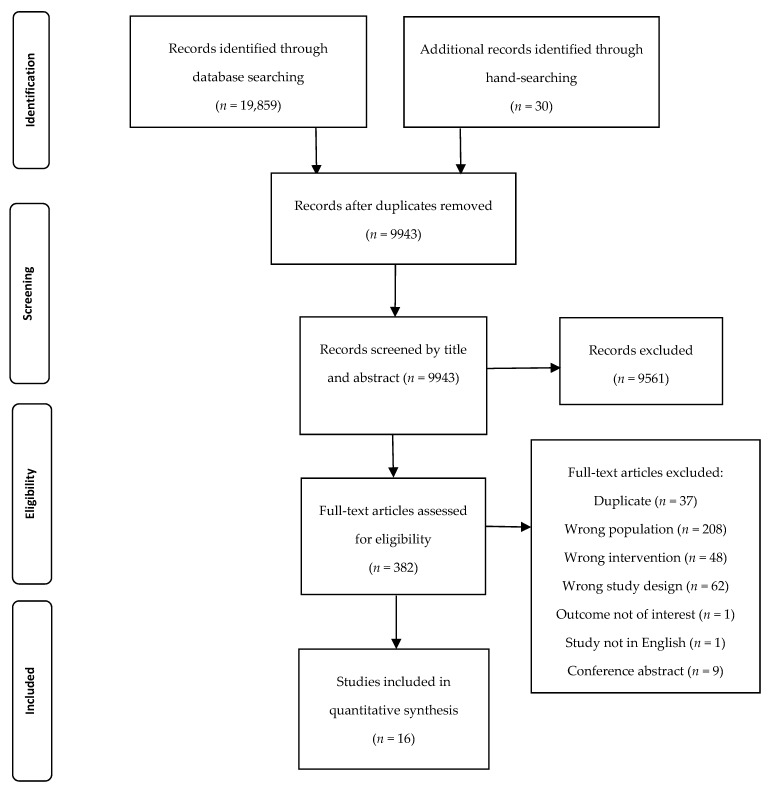
PRISMA flow diagram of included studies and reasons for exclusion at each screening stage.

**Table 1 nutrients-12-00299-t001:** Eligibility criteria for included studies.

	Inclusion	Exclusion
Population	Apparently healthy adults Aged ≥35 years	Participants with an objectively diagnosable disease Trained athletes Results including participants aged <35 years
Intervention	Predominantly whey or colostrum based, high protein milk formula Consumed in beverage form	Whey protein in the form of snack food Colostrum or whey in capsule form Non-specific milk protein, “milk protein isolate” or “milk-based protein” (not primarily whey or colostrum) Addition of creatine to beverage
Comparator	Not specified, but must be present	
Outcome	Objectively measured anthropometric and/or biomarkers related to human health	Subjective measurements Post-prandial studies (≤24 h)
Study design	Randomised controlled trials	Observational studies
Language	English	Languages other than English

**Table 2 nutrients-12-00299-t002:** Bone mineral density related outcomes: results and study details of the effect of milk basic protein, whey protein isolate or colostrum, on bone mineral density of healthy adults aged ≥35 years, in ascending dose of protein.

	Study Details							Results		
**Whey**	Author (Country)	Intervention #	Control # §	Dropout rate (%)	Includes RT	Duration	Sex	Bone mineral density	Biochemistry (relating to BMD)	Urinary markers
	Aoe [38] 2005 (Japan)	MBP 40 mg/day (40 mg protein/day) *n* = 14; 50 ± 3 years *	“matching placebo beverage” *n =* 13; 51 ± 3 years *	16	No	6 months	F	↔ (Δ = 1.87%, CV = 2.0%)	Osteocalcin ↔	NTx ↓
	Kerstetter [42] 2015 (USA)	WPI 45 g/day (40 g protein/day) *n =* 61; 69.9 ± 6.1 years	Maltodextrin *n =* 60; 70.5 ± 6.4 years	24	No	18 months	85% F	↔	PTH ↔ 9 months CTX and P1NP ↑ 18 months CTX ↑ and P1NP ↔ Osteocalcin ↔ 18 months IGF-1 ↑	24 hr urinary urea ↑ 9 months eGFR ↑ UCa ↑ 18 months eGFR ↔ UCa ↔
**Colostrum**	Duff [40] 2014 (Canada)	Colostrum 60 g/day (38 g protein/day) *n =* 19; 61.8 ± 4.8 years	Whey protein complex 60 g/day (38 g protein/day) *n =* 20; 57.5 ± 6.3 years	3	Yes, both groups	8 weeks	64% F	↑ NDBG	NA	Ntx colostrum ↓ whey ↔

CTX, C-terminal telopeptide; CV, coefficient of variation; eGFR, estimated glomerular filtration rate; F, female; IGF-1, insulin-like growth factor; MBP, milk basic protein; NA, not assessed; NDBG, no difference between groups; Ntx, cross-linked n-telopeptides of type I collagen; P1NP, procollagen type 1 N-terminal propeptide; PRISE, multimodal exercise program; RT, resistance training; UCa, urinary calcium; USA, United States of America; WPI, whey protein isolate; Δ, change; ↓, decreased; ↑, increased; ↔, no change. # Daily intake, unless specified. Could be across multiple doses, e.g., three daily doses of 20 g = 60 g total. § Isocaloric to intervention, unless specified. * All such values are mean age ± standard deviation.

**Table 3 nutrients-12-00299-t003:** Body composition outcomes: results and study details of the effect of non-specific whey, whey protein isolate, whey protein concentrate or colostrum, on body composition outcomes of healthy adults aged ≥35 years, grouped according to co-interventions and ordered in ascending dose of protein.

Study Details								Results				
**Whey only**	Author (Country)	Intervention #	Control # §	Dropout rate (%)	Includes RT	Duration	Sex	Body mass Ŧ	Lean tissue mass Ŧ	Fat mass Ŧ	Waist	Hip
	Norton [37] 2016 (Ireland)	Whey blend (“milk based protein matrix”) 0.165 g protein(kg·day) (~12 g protein/day average) *n =* 31; 62.2 ± 4.7 years *	Maltodextrin *n =* 29; 59.5 ± 5.8 years *	51	No	24 weeks	77% F	↑ Whey 0.70 kg (95% CI 0.01, 0.35) Control -0.22 kg (95% CI -0.79, 0.35) Not sig. when expressed as %	↑ Whey 0.45 kg (95% CI 0.06, 0.83) Control -0.16 kg (95% CI -0.49, 0.17) Not sig. when expressed as %	↔	NA	NA
	Kerstetter [42] 2015 (USA)	WPI 45 g/day (40 g protein/day) *n =* 61; 69.9 ± 6.1 years	Maltodextrin *n =* 60; 70.5 ± 6.4 years	24	No	18 months	85% F	↔	9 months ↔ 18 months ↔ (↓ in CHO group)	9 months ↔ 18 months ↔ (↑ in CHO group)	NA	NA
	Pal [28] 2010b (Australia) (same study as Pal [27] 2010a)	WPI 60 g/day (54 g protein/day) *n =* 25; All groups; 48·4 years (SEM 0·86)	Glucose *n =* 25 Casein *n =* 20	21	No	12 weeks	86% F	↔	↔	↔	↔	Waist: hip ratio ↔
	Baer [32] 2011 (USA)	WPC 102 g/day (55 g protein/day) *n =* 23; 49 ± 9 years	Maltodextrin *n =* 25; 51 ± 9 years SPI *n =* 25; 53 ± 9 years	19	No	23 weeks	54% F	↓ vs. CHO ↔ vs. Soy By BodPod^Ⓡ^	↔ By BodPod^Ⓡ^	↓ vs. CHO ↔ vs. Soy By BodPod^Ⓡ^	↓	↔
**Whey and energy deficit diet**	Frestedt [35] 2008 (USA)	Whey & peptides (20 g protein/day) *n =* 31; 43.6 ± 1.1 years	Maltodextrin *n =* 28; 42.0 ± 1.2 years	44	No	12 weeks	F	↓ NDBG	↓ NDBG	↓ (more than control)	↓ DBG	↓ NDBG
	Hector [36] 2015 (Canada)	WPI (54 g protein/day) *n =* 14; 52 ± 2 years	Maltodextrin *n =* 12; 48 ± 3 years SPI *n =* 14; 52 ± 2 years	3	No	14 days	53% F	↓ NDBG	↓ NDBG	↓ NDBG	NA	NA
**Whey and resistance training**	Francis [34] 2017 (Ireland)	Whey blend (“milk based protein matrix”) 0.33 g protein(kg·day) (~23 g/day average) *n =* 28; 61.8 ± 4.5 years	Whey blend + RT *n =* 29; 60.4 ± 5.6 years	42	Yes, in control group only	12 weeks	F	Not reported	↓ Whole body ↑ Upper leg * sig not reported	Not reported	NA	NA
	Eliot [29] 2008 (USA) (same study as Bemben [30])	Whey + Gatorade™ (35 g protein/day, 3× week) Not isocaloric to controls *n =* 11; 58.2 ± 2 years	Gatorade™ *n =* 10; 56.1 ± 1.4 years Gatorade™ + Creatine (5 g creatinine/day, 3× week) *n =* 10; 56.1 ± 1.8 years Gatorade™, whey + creatine (35 g whey + 5 g creatinine/day, 3× week) *n =* 11; 57.2 ± 2.2 years	Not reported	Yes, all groups	14 weeks	M	↔	↑ (no difference between protein groups)	↔	NA	NA
	Chalé [41] 2013 (USA)	WPC (40 g protein/day) *n =* 28; 78 ± 4 years	Maltodextrin *n =* 31; 77.3 ± 3.9 years	15	Yes, both groups	6 months	59% F	↑ NDBG	↑ NDBG (Δ = 1.3%, CV = <4%)	↔ NDBG	NA	NA
	Arciero [31] 2014 (USA)	Whey (60 g protein/day) *n =* 18; 50 ± 2 years	Whey + RT (60 g protein/day) *n =* 22; 47 ± 1 years Whey + PRISE (60 g protein/day) *n =* 17; 52 ± 1 years	28	Yes, in control groups only	16 weeks	63% F	↓ NDBG	↓ NDBG (Δ = 0.6%, CV = 0.64%)	↓ NDBG (Δ = 0.6%, CV = 2.2%) SAT ↓ NDBG VAT ↔	↓ NDBG	NA
**Colostrum**	Duff [40] 2014 (Canada)	Colostrum 60 g/day (38 g protein/day) *n =* 19; 61.8 ± 4.8 years	Whey protein complex 60 g/day (38 g protein/day) *n =* 20; 57.5 ± 6.3 years	3	Yes, both groups	8 weeks	64% F	↔	↑ NDBG (Δ = not reported, CV = 0.5%)	↔ % fat ↓ NDBG (Δ = not reported, CV = 3.0%)	NA	NA

CHO, carbohydrate; CV, coefficient of variation; F, female; M, male; n, number of participants that completed the study; NA, not assessed; NDBG, no difference between groups; PRISE, multimodal exercise program; RT, resistance training; SAT, subcutaneous adipose tissue; SPI, soy protein isolate; USA, United States of America; VAT, visceral adipose tissue; WPC, whey protein concentrate; WPI, whey protein isolate; Δ, change; ↓, decreased; ↑, increased; ↔, no change. # Daily intake, unless specified. Could be across multiple doses, e.g., three daily doses of 20 g = 60 g total. § Isocaloric to intervention, unless specified. * All such values are mean age ± standard deviation. Ŧ Measured by DXA, unless specified.

**Table 4 nutrients-12-00299-t004:** Muscle synthesis and strength outcomes: results and study details of the effect of non-specific whey, whey protein isolate, whey protein concentrate or colostrum, on muscle synthesis and strength outcomes of healthy adults aged ≥35 years, grouped according to outcomes measured and ordered in ascending dose of protein.

	Study Details							Results		
	Author Year	Intervention #	Control # §	Dropout rate (%)	Includes RT	Duration	Sex	Muscle synthesis	Strength	Other
**Whey**	Francis [34] 2017 (Ireland)	Whey blend (“milk based protein matrix”) 0.33 g protein(kg·day) (~23 g average/day) *n =* 28; 61.8 ± 4.5 years *	Whey blend + RT *n =* 29; 60.4 ± 5.6 years	42	Yes, in control group only	12 weeks	F	NA	Knee extensor torque ↓ Muscle quality ↓ 900 m gait speed ↓ Chair rises ↓ * sig not reported	NA
	Bemben [30] 2010 (USA) (same study as Eliot [29])	Whey + Gatorade™ (35 g protein/day, 3× week) Not isocaloric to controls *n =* 11; 58.2 ± 2 years	Gatorade™ *n =* 10; 56.1 ± 1.4 years Gatorade™ + Creatine (5 g creatinine/day, 3× week) *n =* 10; 56.1 ± 1.8 years Gatorade™, whey + creatine (35 g whey + 5 g creatinine/day, 3× week) *n =* 11; 57.2 ± 2.2 years	Not reported	Yes, all groups	14 weeks	M	NA	Knee extension ↑ (higher in protein groups) (Δ = 30–63%, CV = <1%) All other measures ↑ NDBG (bench press, military press, lat pull, biceps curl, triceps extension, leg press, knee flexion)	NA
	Chalé [41] 2013 (USA)	WPC (40 g protein/day) *n =* 28; 78 ± 4 years	Maltodextrin *n =* 31; 77.3 ± 3.9 years	15	Yes, both groups	6 months	59% F	NA	↑ NDBG (leg press, knee extension, stair-climb, chair-rise) 400 m walk ↔	NA
	Farnfield [33] 2012 (Australia)	WPI (27 g protein/day, 3× week) *n =* 9; 68.1 ± 1.6 years	Placebo (not stated if isocaloric) *n =* 9; 67.4 ± 1.3 years	Unclear	Yes, both groups	12 weeks	M	Signalling markers of MPS ↓ (p70S6K, mTOR, rpS6, eIF4G)	↑ NDBG (leg press, leg extension, bench press)	NA
	Devries [39] 2018 (Canada)	WPI (25 g protein/day) *n =* 11; 68 ± 1 years	Leucine (10 g protein/day containing 3 g leucine) *n =* 10; 69 ± 1 years	4	Yes, both groups	6 days	F	FSR rested leg ↑ (vs. ↔ in leucine group) FSR exercised leg ↑ (but less than leucine group)	NA	NA
	Hector [36] 2015 (Canada)	WPI (54 g protein/day) *n =* 14; 52 ± 2 years Energy deficit diet	Maltodextrin *n =* 12; 48 ± 3 years SPI *n =* 14; 52 ± 2 years Energy deficit diet	3	No	14 days	53% F	MPS ↓ 9% (but less reduced than soy (↓ 28%, *P* = 0.021) or CHO (↓ 31%, P = 0.013)) FSR ↓ (sig not reported)	NA	NA
**Colostrum**	Duff [40] 2014 (Canada)	Colostrum 60 g/day (38 g protein/day) *n =* 19; 61.8 ± 4.8 years	Whey protein complex 60 g/day (38 g protein/day) *n =* 20; 57.5 ± 6.3 years	3	Yes, both groups	8 weeks	64% F	Muscle thickness of knee extensors and elbow flexors ↑ NDBG (Δ = not reported, CV = 2.5%)	Leg press strength ↑ in both, higher in colostrum group (Colostrum Δ = 21%, Whey Δ = 5%, CV = 3.0%) Bench press ↑ NDBG	Cognition ↑ NDBG

CHO, carbohydrate; CV, coefficient of variation; F, female; FSR, fractional synthetic rate; M, male; MPS, myofibrillar protein synthesis; n, number of participants that completed the study; NA, not assessed; NDBG, no difference between groups; RT, resistance training; SPI, soy protein isolate; USA, United States of America; WPI, whey protein isolate; WPC, whey protein concentrate; Δ, change; ↓, decreased; ↑, increased; ↔, no change # Daily intake, unless specified. Could be across multiple doses, e.g. three daily doses of 20g = 60g total § Isocaloric to intervention, unless specified * All such values are mean age ± standard deviation in years.

**Table 5 nutrients-12-00299-t005:** Biochemical and vascular markers: results and study details of the effect of non-specific whey, whey protein isolate, whey protein concentrate or colostrum, on biochemical and vascular markers of healthy adults aged ≥35 years, grouped according to outcomes measured.

Study Details								Results				
	Author (Location)	Intervention #	Control # §	Dropout Rate (%)	Includes RT	Duration	Sex	Glucose	Insulin	Lipids	Vascular Measures	Biochemical Variables
**Whey**	Baer [32] 2011 (USA)	WPC 102 g/day (55 g protein/day) *n =* 23; 49 ± 9 years *	Maltodextrin *n =* 25; 51 ± 9 years SPI *n =* 25; 53 ± 9 years	19	No	23 weeks	54% F	↔ (fasting)	↓ vs. CHO ↔ vs. Soy (fasting)	NA	NA	Ghrelin ↓ IGF-I ↓ vs. soy, ↔ vs. CHO IGFBP-1 ↔ IGFBP-3 ↓ T4 (free) ↓ vs. soy, ↔ vs. CHO T3 (uptake) ↓ vs. soy, ↔ vs. CHO
	Hector [36] 2015 (Canada)	WPI (54 g protein/day) *n =* 14; 52 ± 2 years Energy deficit diet	Maltodextrin *n =* 12; 48 ± 3 years SPI *n =* 14; 52 ± 2 years Energy deficit diet	3	No	14 days	53% F	↔ (post prandial)	↔ (post prandial)	NA	NA	Glycerol Ra ↔
	Pal [28] 2010b (Australia) (same study as Pal [27] 2010a)	WPI 60 g/day (54 g protein/day) *n =* 25; All groups; 48·4 years (SEM 0·86)	Glucose *n =* 25 Casein *n =* 20	21	No	12 weeks	86% F	↔ (fasting)	HOMA-IR ↓ Insulin ↓ (fasting)	TAG ↓ HDL cholesterol ↔ Chol ↓ LDL cholesterol ↓ (fasting)	NA	NA
	Frestedt [35] 2008 (USA)	Whey & peptides (20 g protein/day) *n =* 31; 43.6 ± 1.1 years Energy deficit diet	Maltodextrin *n =* 28; 42.0 ± 1.2 years Energy deficit diet	44	No	12 weeks	F	NA	Measured but not reported	TAG ↔ HDL cholesterol ↔ Chol ↓ LDL cholesterol ↔ (not reported if postprandial or fasting)	NA	NA
	Arciero [31] 2014 (USA)	Whey (60 g protein/day) *n =* 18; 50 ± 2 years	Whey + RT (60 g protein/day) *n =* 22; 47 ± 1 years Whey + PRISE (60 g protein/day) *n =* 17; 52 ± 1 years	28	Yes, in control groups only	16 weeks	63% F	↔ (fasting)	HOMA-IR ↔ Insulin ↔ (fasting)	TAG ↓ HDL cholesterol ↓ Chol ↔ LDL cholesterol ↔ (fasting)	Heart rate ↔ Blood pressure ↔	Leptin ↔ Adiponectin ↔
	Pal [27] 2010a (Australi) (same study as Pal [28] 2010b)	WPI 60 g/day (54 g protein/day) *n =* 25; All groups; 48·4 years (SEM 0·86)	Glucose *n =* 25 Casein *n =* 20	21	No	12 weeks	86% F	NA	NA	NA	SBP ↓ vs. control, ↔ vs. casein DBP ↓ vs. control, ↔ vs. casein Augmentation index ↓	IL-6, CRP, and TNF-α ↔
**Colostrum**	Duff [40] 2014 (Canada)	Colostrum 60 g/day (38 g protein/day) *n =* 19; 61.8 ± 4.8 years	Whey protein complex 60 g/day (38 g protein/day) *n =* 20; 57.5 ± 6.3 years	3	Yes, both groups	8 weeks	64% F	NA	NA	NA	NA	IGF-1 and CRP ↔

CHO, carbohydrate; Chol, cholesterol; CRP, c-reactive protein; DBP, diastolic blood pressure; F, female; HDL, high density lipoprotein; HOMA-IR, homeostatic model assessment-insulin resistance; IGF-1, insulin-like growth factor 1; IGFBP-1, insulin-like growth factor binding protein 1; IGFBP-3, insulin-like growth factor binding protein 3; IL-6, interleukin 6; LDL, low density lipoprotein; M, male; n, number of participants that completed the study; NA, not assessed; PRISE, multimodal exercise program; RT, resistance training; SEM, standard error of the mean; SBP, systolic blood pressure; T3, triiodothyronine; T4, thyroxine; TAG, triacylglycerol; TNF-α, tumour necrosis factor alpha; USA, United States of America; WPC, whey protein concentrate; WPI, whey protein isolate; ↓, decreased; ↑, increased; ↔, no change. # Daily intake, unless specified. May be across multiple doses, e.g., three daily doses of 20 g = 60 g total. § Isocaloric to intervention, unless specified. * All such values are mean age ± standard deviation.

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
