# Peer review of "Health Benefits of Whey or Colostrum Supplementation in Adults ≥35 Years; a Systematic Review"

_nutrients, 2020, doi:10.3390/nu12020299_

Round 1
Reviewer 1 Report
The authors have submitted a systematic literature that covers a range of outcomes related to supplementation (as a beverage) with a variety of whey-derived protein ingredients and colostrum in adults over the age of 35 years of age. Aside from the results, it is very positive to see the collaboration between academia and industry to summarize the current literature on specific topics, which could further foster claims on foods that provide health benefits to the public as well as efforts to address knowledge gaps in the area. Overall, the literature review was well-searched and thorough. The manuscript is well written in a clear and concise manner. Prior to publication it is suggested that some relatively minor issues be addressed by the authors.
Would like more clarification on why 35 years was set as a target population. There are younger adults demonstrating impaired glucose intolerance and diabetes, and perhaps could demonstrate more potency with respect to stimulating muscle protein synthesis. Furthermore, if a health claim were to be made from results from this manuscript, would the claim be required to qualify the age to which the review included?
These questions do not take away from the review, but perhaps could be further flushed out in the methods (rationale for the age) and, perhaps, the 35 year cutoff could be discussed briefly in the Discussion section.
In the results section the numbering system is confusion. After “3.1.3 Body Composition” the section “whey supplementation alone” have no numbers. Also, how does the latter differ from “3.1 whey supplementation.” Then there is “1 whey and energy deficit,” “2 whey and resistance training” etc. Unsure if “1” and “2.” Are part of “whey supplementation alone” section?
Are the sections “muscle strength” and “muscle synthesis” part of the “whey supplementation alone” section?
In the same section for “whey and energy deficit” and “whey and resistance training” it is suggested that you replace “and” with “+.” This makes it clearer to the reader that whey was provided in addition to an imposed energy deficit and resistance training. It is also suggested that in the corresponding paragraphs you briefly describe that this was the intervention. This was already done for in the first section of the “whey and resistance training” paragraph.
In the Discussion section, it could be helpful to briefly differentiate between general and high level health claims in Australia. It is my understanding that high level health claims that refer to a disease biomarker do indeed require assessment by FSANZ? Section 1.2.7-18 of the Food Standards Code refers to submitting for high level and general health claims. In section 2, high level claims must be in the corresponding table from Schedule 4 Table 5. A general health claim (section 3) can be from the Table in Schedule 4 or the petitioner must notify FSANZ that the relationship has been established by a systematic review. Please confirm this your rebuttal or briefly clarify in the discussion section.
Reviewer 2 Report
This is a nice systematic review in an important area. I think that upon some small improvements it is suitable for publication. I am confident that the outcome reflects the state of art in this area. Below a list of issues for the authors to consider, essentially in order of line appearance.
Line 12: ‘may exert’: given the results of the work this is too strong a statement. “are thought to”? “are sometimes claimed to”? or another softer statement Line 39: European Food SAFETY Authority Line 39: ref 6 is OK. And there are quite a few journal papers that also illustrate the state of claims in EU. When added, those can also be informative in addition to a website. Line 40: there are 3 levels of outcome in EFSA (not 2): “substantiated, insufficient evidence, not substantiated”. Only the first can lead to EU decisions to allow a health claim. Authors themselves also have “insufficient” in line 321 and “inconclusive” in line 369. And “denied” and “approved” is not the terminology used by EFSA, so I would not use these in reference to EFSA. Table 1 and full paper (eg line 148, 149). Authors could (should?) consider to exclude all non-protein controls. I am pretty sure that EFSA would only consider protein controls as adequate. In Table 1 then carbohydrate control are an exclusion criterion. I know this is fundamental. Line 284/285: very true. Table 2 etc. These are containing the info but not easily accessible. Authors can consider to (also) present info in graphs, eg study duration, amount tested etc. Line 332: EFSA/EU model Liens 329-343: also refer to the EU register of claims (https://ec.europa.eu/food/safety/labelling_nutrition/claims/register/public/?event=register.home ) and the associated Regulation 432/2012.
Final relevant remark in general: Authors consider little on characterisation. Yet EFSA considers adequate characterisation of the intervention materials as the pivotal first step. This could (should?) be an additional consideration/criterion (and even more studies may drop out)
